# Why Popular MOEAs Are Popular: Proven Advantages in Approximating the Pareto Front

**Mingfeng Li**[1][*],  **Qiang Zhang**[1][*],  **Weijie Zheng**[1,2][†],  **Benjamin Doerr**[3]

[1]School of Computer Science and Technology,
State Key Laboratory of Smart Farm Technologies and Systems,
International Research Institute for Artificial Intelligence,
Harbin Institute of Technology, Shenzhen, China
[2]Pengcheng Laboratory, Shenzhen, China
[3]Laboratoire d'Informatique (LIX), CNRS, École Polytechnique,
Institut Polytechnique de Paris, Palaiseau, France
limingfeng@stu.hit.edu.cn, zhengweijie@hit.edu.cn, doerr@lix.polytechnique.fr

## Abstract

Recent breakthroughs in the analysis of multi-objective evolutionary algorithms (MOEAs) are mathematical runtime analyses of those algorithms which are intensively used in practice. So far, most of these results show the same performance as previously known for simpler algorithms like the GSEMO. The few results indicating advantages of the popular MOEAs share the same shortages: They only consider the problem of computing the full Pareto front, sometimes of algorithms enriched with newly invented mechanisms, and this on newly designed benchmarks. In this work, we overcome these shortcomings by analyzing how existing popular MOEAs approximate the Pareto front of the established LARGEFRONT benchmark. We prove that several popular MOEAs, including NSGA-II (with current crowding distance), NSGA-III, SMS-EMOA, and SPEA2, only need an expected time of $O(n^2 \log n)$ fitness evaluations to compute an additive $\varepsilon$-approximation of the Pareto front of the LARGEFRONT benchmark. This contrasts with the already proven exponential runtime (with high probability) of the GSEMO on the same task. Our result is the first mathematical runtime analysis showing and explaining the superiority of popular MOEAs over simple ones like the GSEMO for the central task of computing good approximations to the Pareto front.

## 1 Introduction

Mathematical runtime analyses of the evolutionary algorithms are usually challenging. Early theoretical results for multi-objective evolutionary algorithms (MOEAs) [LTZ+02, LTZ04] focused on the simple algorithms like the (G)SEMO, which use only dominance for the survival selection. Recent breakthroughs like the runtime analysis on the most widely used MOEA, NSGA-II [DPAM02], successfully conducted in [ZLD22, ZD23], have triggered a new era in the theory of MOEAs. Other algorithms that are intensively used in practice, like the NSGA-III [DJ14], SMS-EMOA [BNE07], and SPEA2 [ZLT01], were theoretically analyzed soon thereafter [WD23, BZLQ23, RBLQ24], letting the theory of these popular MOEAs quickly become a hot topic [BQ22, DQ23a, DOSS23b, BZLQ23, DDHW23, WD23, ZD24b, ZLDD24, DZL+24, ZD24a, ODNS24, RBLQ24, DIK25, DZD25, LZD25].

---

[*]Equal Contribution.
[†]Corresponding author.

39th Conference on Neural Information Processing Systems (NeurIPS 2025).

Interestingly, despite the rapid progress in the analysis of practical MOEAs, only very few results have demonstrated theoretical advantages of popular MOEAs over simpler algorithms. Dang et al. [DOSS23a] introduced the drastic Bernoulli noise model, and showed that the GSEMO fails badly on every Bernoulli-noisy fitness function, while the NSGA-II can cover the whole Pareto front of the Bernoulli-noisy LOTZ problem in polynomial time. Dang et al. [DOS24] proved that three popular MOEAs, namely NSGA-II, NSGA-III and SMS-EMOA, enhanced with a mild diversity mechanism (avoiding genotype duplication), require $O(n \log n)$ expected fitness evaluations to cover the whole Pareto front of their newly designed ONETRAPZEROTRAP problem, which only has two extremal points as the whole Pareto front. In contrast, the simpler algorithm GSEMO requires at least $n^n$ fitness evaluations in expectation. The very recent work [DOS25] constructed an artificial problem with a small Pareto set where almost all pairs of search points are incomparable, also with only two points in the whole Pareto front, and proved that any black-box MOEA using only dominance-based selection and bit-value-invariant variation operators takes exponential time with high probability, while three popular MOEAs, namely NSGA-II, NSGA-III, and SMS-EMOA, efficiently cover the Pareto front in expected quadratic time.

The above results [DOSS23a, DOS24, DOS25] indicating advantages of popular MOEAs share the same shortages. They consider the performance for the problem of computing the full Pareto front, (of some algorithms enriched with newly invented mechanisms), and this on newly designed benchmarks. In practice, one cannot know the Pareto front beforehand. The newly invented mechanisms or newly designed benchmarks place the question on the generality of tailored results. Till now, it is still not convincingly proved in theory why popular MOEAs are popular in practice.

**Our contributions**: This work undertakes an attempt to overcome these shortages by analyzing how several popular MOEAs (NSGA-II, NSGA-III, SMS-EMOA, and SPEA2) approximate the Pareto front of the LARGEFRONT$'_\varepsilon$ benchmark (denoted by LF$'_\varepsilon$) proposed in [HN09]. Note that we do not consider MOEA/D here, since it is structurally very different from the domination-based algorithms analyzed in this work and poses additional challenges due to its decomposition mechanism. We prove that, for LF$'_\varepsilon$ with problem size $n$, these four popular MOEAs achieve an additive $\varepsilon$-approximation of LF$'_\varepsilon$ better in an expected number of $O(n^2 \log n)$ fitness evaluations (see Theorems 9, 12, 14 and 16). In contrast, an existing result from [HN09] showed that the GSEMO fails to accomplish this task in expected polynomial time (see Theorem 5). We also provide a general theorem showing an expected runtime of $O(\mu n \log n)$ fitness evaluations for finding an additive $\varepsilon$-approximation of LF$'_\varepsilon$ for any MOEA with population size at most $\mu$ satisfying a general property of the selection of the next population (see Theorem 7). Compared with the GSEMO, which only applies the dominance criterion for survival selection, these popular MOEAs additionally use a criterion to increase the diversity of the survive individuals in the next population. This will result in a better approximation when the number of Pareto front points is large. This provides the first mathematical runtime analysis showing the superiority of popular MOEAs over simpler ones like the GSEMO for the central task of computing good approximations to the Pareto front.

The rest of the paper is organized as follows. Section 2 defines the approximation measure and the known LARGEFRONT$_\varepsilon$ benchmark. Section 3 presents a general approximation theorem, and Section 4 applies it to establish runtime guarantees for NSGA-II, NSGA-III, SMS-EMOA, and SPEA2 for computing good approximations. Section 5 concludes our paper.

## 2 Preliminaries

### 2.1 Additive $\varepsilon$-Approximation

We first recall some basic definitions for the maximization of a bi-objective problem $f = (f_1, f_2) : \Omega \to \mathbb{R}^2$ defined on the search space $\Omega$. For $x, y \in \Omega$, we say that $x$ *weakly dominates* $y$, written as $x \succeq y$, if $f_1(x) \geq f_1(y)$ and that $f_2(x) \geq f_2(y)$, and $x$ *dominates* $y$, written as $x \succ y$, if in addition at least one inequality is strict. A solution $x \in \Omega$ is a *Pareto optimum* if no other solution dominates it. The *Pareto set* consists of all Pareto optima. The set of corresponding objective values is called the *Pareto front*.

When the Pareto front is excessively large or infinite, covering the whole Pareto front is infeasible and a good approximation of the Pareto front becomes a natural goal. There are multiple approximation measures, such as $\varepsilon$-dominance [LTDZ02], generational distances [VVL98, BT03, CCRS04], hyper-volume indicator [ZT98] or maximal empty interval size [ZD25]. Here we adhere to the original

LARGEFRONT$'_\varepsilon$ work [HN09], and use additive $\varepsilon$-approximation (see Definition 1) to evaluate how well a set of points approximates the Pareto front. It is built on the additive $\varepsilon$-dominance, first defined in [LTDZ02], that relaxes the usual dominance relation by allowing an additive slack $\varepsilon$ in each objective.

**Definition 1** ([LTDZ02]). *A set of objective vectors $T$ is an additive $\varepsilon$-approximation of $f$ : $\{0,1\}^n \to \mathbb{R}^m$ if and only if for each objective vector $v \in f(\{0,1\}^n)$, there exists at least one objective vector $u \in T$ that additively $\varepsilon$-dominates $v$, where an objective vector $u$ is said to additively $\varepsilon$-dominate $v$ (written as $u \succeq_\varepsilon v$) if and only if $u_i + \varepsilon \ge v_i$ for all $i \in \{1, \dots, m\}$.*

### 2.2 The LargeFront Benchmark

LARGEFRONT$_\varepsilon$ is a benchmark proposed in [HN08] and [HN09]. It exists in two variants, LF$_\varepsilon$ [HN08] and LF$'_\varepsilon$ [HN09]. Different from existing benchmarks for theoretical analysis, like COCZ [LTZ04], LOTZ [LTZ04], ONEMINMAX [GL10], OJZJ [DZ21], which have the polynomial Pareto fronts, for both variants the Pareto fronts have exponential size (see Lemma 3). Since LF$'_\varepsilon$ shows more similarity to the arguably most popular ONEMINMAX benchmark, this paper will only discuss this variant. We hope that our findings will inspire analyses of MOEAs on the other variant LF$_\varepsilon$. Following is the definition of LF$'_\varepsilon$.

**Definition 2** ([HN09]). *Let $n \in \mathbb{N}$ be even and $\varepsilon > 0$. The function $LF'_\varepsilon(x) = (LF'_{\varepsilon,1}(x), LF'_{\varepsilon,2}(x))$ : $\{0,1\}^n \to \mathbb{R}^2$ is defined by*

$$LF'_{\varepsilon,1}(x) := \begin{cases} \left(2|x'|_1 + 2^{-n/2}BV(x'')\right) \cdot \varepsilon & \min\{|x'|_1, |x'|_0\} \ge \sqrt{n} \\ 2|x'|_1 \cdot \varepsilon & \text{otherwise} \end{cases}$$

$$LF'_{\varepsilon,2}(x) := \begin{cases} \left(2|x'|_0 + 2^{-n/2}BV(\overline{x''})\right) \cdot \varepsilon & \min\{|x'|_1, |x'|_0\} \ge \sqrt{n} \\ 2|x'|_0 \cdot \varepsilon & \text{otherwise,} \end{cases}$$

*where $x'$ and $x''$ are the prefix and suffix of length $n/2$ of $x$, $|\cdot|_1$ and $|\cdot|_0$ denote the number of ones and the number of zeros in this bitstring respectively, and $BV(y) : \{0,1\}^{n'} \to \mathbb{R}$ is defined by $BV(y) = \sum_{i=1}^{n'} 2^{n'-i} y_i$, computing the decimal value of the $n'$-bit binary number $y$.*

An illustration of this function is given in Figure 1.

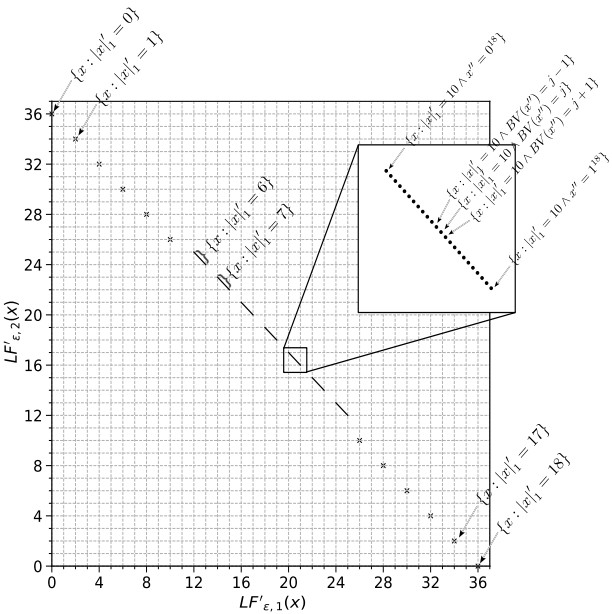

Figure 1: Objective space of $LF'_\varepsilon$ for $\varepsilon = 1$ and $n = 36$.

As stated in [HN09], all bitstrings are Pareto optimal. Since any two solutions $x$ and $y$ such that $x'' \ne y''$ have different objective values, the size of the Pareto front grows exponentially with the

problem size $n$. The following lemma collects the results of the Pareto set and the Pareto front w.r.t. $\mathrm{LF}'_\varepsilon$. Note that we use the notation of $[a..b] := \{a, a+1, \ldots, b\}$ for $a \leq b$ and $a, b \in \mathbb{Z}$.

**Lemma 3** ([HN09])**.** *The Pareto set of* $\mathrm{LF}'_\varepsilon$ *is* $S^* = \{0, 1\}^n$*, that is, every bitstring of length* $n$ *is Pareto optimal. The Pareto front is* $F^* = \{((2k + 2^{-n/2}\alpha)\varepsilon, (n - 2k + 2^{-n/2}(2^{n/2} - 1 - \alpha))\varepsilon) \mid k \in [0..n/2], \alpha \in [0..2^{n/2} - 1]\}$*, where* $\alpha = 0$ *for* $k < \sqrt{n}$ *or* $k > n/2 - \sqrt{n}$*, and for* $\sqrt{n} \leq k \leq n/2 - \sqrt{n}$*,* $\alpha$ *ranges over all integers in* $[0..2^{n/2} - 1]$*.*

The following lemma from [HN09] gives a necessary and sufficient condition for a set to be an additive $\varepsilon$-approximation w.r.t. $LF'_\varepsilon$.

**Lemma 4** ([HN09])**.** *A set* $T$ *is an additive* $\varepsilon$-*approximation of* $LF'_\varepsilon$ *if and only if for each* $k \in \{0, \ldots, n/2\}$ *there exists a solution* $x \in T$ *with* $|x'|_1 = k$ *.*

As common in the evolutionary computation theory community [NW10, AD11, Jan13, ZYQ19, DN20], by *runtime* we usually mean the number of fitness evaluations to reach a specific goal. Horoba and Neumann [HN09] proved that the GSEMO fails to achieve an additive $\varepsilon$-approximation of $\mathrm{LF}'_\varepsilon$ in polynomial runtime.

**Theorem 5** ([HN09])**.** *The runtime until the GSEMO has achieved an additive* $\varepsilon$-*approximation of* $\mathrm{LF}'_\varepsilon$ *is* $2^{\Omega(n^{1/4})}$ *with probability* $1 - 2^{-\Omega(n^{1/4})}$*.*

As stated above, in this work, we aim to analyze the runtime of popular MOEAs to achieve an additive $\varepsilon$-approximation of $\mathrm{LF}'_\varepsilon$.

## 3 General Approximation Theorem

Before proving runtime results specific to a popular MOEA, this section will formulate a general theorem asserting a runtime guarantee (to reach an additive $\varepsilon$-approximation w.r.t. $\mathrm{LF}'_\varepsilon$) for a general MOEA framework. It will be then used to prove runtime guarantees of popular MOEAs in next section. We believe that it will be useful also for future research on $\mathrm{LF}'_\varepsilon$. Algorithm 1 states the general MOEA framework regarded in this section. The population (the set of solutions) is initialized uniformly at random. In each generation, the algorithm chooses the mating population, generates $\lambda$ offspring individuals, and then uses a survival selection to determine the next population. We note that this framework with $\lambda = 1$, $p_c = 0$, random parent selection, bit-wise mutation, and dominance-only survival selection gives the GSEMO. The setting of $\lambda = |P_t|$, the survival selection of non-dominated sorting and crowding distance gives the NSGA-II. The NSGA-III corresponds to $\lambda = |P_t|$, survival selection by non-dominated sorting and reference point mechanism. The setting of $\lambda = |P_t|$, the survival selection of non-dominated sorting and hypervolume contribution indicator gives the SMS-EMOA. The SPEA2 fits this framework by maintaining a fixed population size, generating $\lambda$ offspring per iteration, and employing strength-and-density estimation for survival selection.

---

**Algorithm 1:** A general MOEA framework

---

1  Initialize $P_0$ uniformly at random;
2  **for** $t = 0, 1, 2, \ldots$ **do**
3  $\quad$ Choose $\lambda$ individuals from $P_t$ to form the mating population $P'_t$;
4  $\quad$ Generate the offspring population $Q_t$ with $\lambda$ individuals from $P'_t$ via applying crossover
$\quad\quad$ (with crossover rate $p_c$) and mutation;
5  $\quad$ Select the next population $P_{t+1}$ from $R_t = P_t \cup Q_t$ via a specific survival selection;

---

We define the following property for the survival selection that will be used for our general theorem.

**Definition 6** (Property $\mathcal{A}$)**.** *An MOEA, or more specifically its survival selection procedure, satisfies Property $\mathcal{A}$ on* $\mathrm{LF}'_\varepsilon$ *if whenever at some time* $t$ *the combined population* $R_t = P_t \cup Q_t$ *contains an* $x$ *with* $|x'|_1 = k$*, the next population* $P_{t+1}$ *contains a* $y$ *with* $|y'|_1 = k$*, where* $x'$ $(y')$ *is the first half sub-bitstring of* $x$ $(y)$*.*

Property $\mathcal{A}$ ensures that once a value of $k$ 1-bits in the first half of bitstring is discovered, it will never be lost. Together with Lemma 4, we bound the expected number of fitness evaluations to achieve an

additive $\varepsilon$-approximation of $\mathrm{LF}'_\varepsilon$ by $O(\mu n \log n)$ in the following theorem. Due to the space limit, we omit all our proofs here. They can be found in the extended of this paper on the arXiv preprint server.

**Theorem 7.** *Let the crossover rate $p_c \in [0, 1)$. Let $\mu$ be an upper bound on the size of parent population $P$ with $\mu > n/2$. Let $\lambda$ be the size of offspring population $Q$ with $\lambda = O(\mu)$. Consider using Algorithm 1 with random selection, one-bit mutation or bit-wise mutation to generate $P'$, and survival selection with Property $\mathcal{A}$, to optimize $\mathrm{LF}'_\varepsilon$. Then the expected number of fitness evaluations for achieving an additive $\varepsilon$-approximation is $O(\mu n \log n)$.*

## 4 Approximation Guarantees for Popular MOEAs

Based on the general approximation theorem (Theorem 7) in the previous section, this section will prove $O(n^2 \log n)$ expected runtimes for obtaining an additive $\varepsilon$-approximation w.r.t. $\mathrm{LF}'_\varepsilon$ for four widely used MOEAs, namely NSGA-II, NSGA-III, SMS-EMOA, and SPEA2, all by majorly proving that these popular MOEAs satisfy Property $\mathcal{A}$.

### 4.1 NSGA-II Using the Current Crowding Distance

The Non-dominated Sorting Genetic Algorithm II (NSGA-II [DPAM02]), is the most widely used MOEA in practice. As stated in the Section 1, Zheng et al. [ZLD22, ZD23] conducted the first mathematical runtime analysis of the NSGA-II, inspiring a series of follow-up studies. Among them, only Zheng and Doerr [ZD22, ZD25] analyzed how the NSGA-II approximates the Pareto front. These works suggest that the original NSGA-II has difficulties computing good approximation. In contrast, they also proved that a simple modification, like using the current crowding distance in the survival selection [KD06], or a steady-state mode [DNLA09], will result in a near-ideal approximation of the Pareto front for the ONEMINMAX benchmark. Since the proofs are quite similar for these two variants, this work will only discuss the NSGA-II with the current crowding distance. We conjecture similar results for the steady-state variant.

The NSGA-II (see Algorithm 2) fits into the general MOEA framework (Algorithm 1), with fixed population size $N$, offspring population size $\lambda = N$, and a special survival selection. The survival selection uses the dominance as the first criterion. More precisely, it uses the non-dominated sorting to divide the combined population $R_t$ into several fronts $F_1, F_2, \ldots$ such that $F_i$ consists of the non-dominated individuals of $R_t \setminus \bigcup_{j=1}^{i-1} F_i$. For the critical front $F_{i^*}$ with $\sum_{i=1}^{i^*-1} |F_i| < N \leq \sum_{i=1}^{i^*} |F_i|$, the crowding distance is calculated (see Algorithm 3). The original NSGA-II directly removes $|\bigcup_{i=1}^{i^*} F_i| - N$ individuals with smallest crowding distance in $F_{i^*}$ and selects the remaining ones in $F_{i^*}$. This strategy only uses the initial crowding distance, and ignores the changes of crowding distance of remaining individuals after each removal. Hence, Kukkonen and Deb [KD06] proposed the survival selection with the current crowding distance and Zheng and Doerr [ZD25] proved its approximation advancing against the original one. Since each removal only affects the crowding distance of four individuals, the update of the crowding distance can be effectively implemented [ZD25].

The following lemma shows that the NSGA-II with current crowding distance satisfies Property $\mathcal{A}$ when the population is large enough.

**Lemma 8.** *Let $N \geq \frac{2n}{3} + 3$. Consider using the NSGA-II with the survival selection based on the current crowding distance to optimize $\mathrm{LF}'_\varepsilon$ with problem size $n$. Assume that at some iteration $t$, the combined parent and offspring population $R_t = P_t \cup Q_t$ contains an individual $x$ with $|x'|_1 = k$. Then the next parent population $P_{t+1}$ also contains an individual $y$ with $|y'|_1 = k$.*

With Lemma 8, we easily apply Theorem 7 to the NSGA-II with current crowding distance and obtain an $O(n^2 \log n)$ expected runtime to reach an additive $\varepsilon$-approximation w.r.t. $\mathrm{LF}'_\varepsilon$ when setting $N = \Theta(n)$.

**Theorem 9.** *Let $N \geq \frac{2n}{3} + 3$ and $p_c \in [0, 1)$. Consider using the NSGA-II with the survival selection based on the current crowding distance and employing uniform selection and one-bit mutation or bit-wise mutation to optimize $\mathrm{LF}'_\varepsilon$ with problem size $n$. Then after an expected number of $O(Nn \log n)$ fitness evaluations, the population is an additive $\varepsilon$-approximation w.r.t. $\mathrm{LF}'_\varepsilon$.*

**Algorithm 2:** NSGA-II using current crowding distance [KD06, ZD25]

1  Generate $P_0$ by selecting $N$ solutions uniformly and randomly from $\{0,1\}^n$ with replacement;
2  **for** $t = 0, 1, 2, \ldots$ **do**
3      Generate the offspring population $Q_t$ with size $N$;
4      Use fast-non-dominated-sort() in [DPAM02] to divide $R_t$ into fronts $F_1, F_2, \ldots$;
5      Find $i^* \geq 1$ such that $|\bigcup_{i=1}^{i^*-1} F_i| < N$ and $|\bigcup_{i=1}^{i^*} F_i| \geq N$;
6      Use Algorithm 3 to separately calculate the crowding distance of each individual in $F_1, \ldots, F_{i^*}$;
7      **while** $|\bigcup_{i=1}^{i^*} F_i| \neq N$ **do**
8          Let $x$ be the individual with the smallest crowding distance in $F_{i^*}$, chosen at random in case of a tie;
9          Find four neighbors of $x$, two in the sorted list with respect to $f_1$ and two for $f_2$. Update the crowding distance of these four neighbors;
10         $F_{i^*} = F_{i^*} \setminus \{x\}$;
11     $P_{t+1} = \bigcup_{i=1}^{i^*} F_i$

---

**Algorithm 3:** Computation of the crowding distance $\mathrm{cDis}(S)$ [DPAM02]

**Input:** $S = \{S_1, S_2, \ldots, S_{|S|}\}$
**Output:** $\mathrm{cDis}(S) = (\mathrm{cDis}(S_1), \mathrm{cDis}(S_2), \ldots, \mathrm{cDis}(S_{|S|}))$, where $\mathrm{cDis}(S_i)$ is the crowding distance for $S_i$
1  $\mathrm{cDis}(S) = (0, \ldots, 0)$;
2  **for** each objective $f_i$ **do**
3      Sort $S$ in order of descending $f_i$ value: $S_{i.1}, \ldots, S_{i.|S|}$;
4      $\mathrm{cDis}(S_{i.1}) = +\infty$, $\mathrm{cDis}(S_{i.|S|}) = +\infty$;
5      **for** $j = 2, \ldots, |S| - 1$ **do**
6          $\mathrm{cDis}(S_{i.j}) = \mathrm{cDis}(S_{i.j}) + \dfrac{f_i(S_{i.j-1}) - f_i(S_{i.j+1})}{f_i(S_{i.1}) - f_i(S_{i.|S|})}$;

---

### 4.2 NSGA-III

The NSGA-II was reported to encounter difficulties for problems with more objectives (and recently it was proven that at least an exponential runtime is needed to cover the full Pareto front for $m$ONEMINMAX, with three and more objectives [ZD24a]). Deb and Jain [DJ14] proposed a new variant called the Non-dominated Sorting Genetic Algorithm III, NSGA-III, to overcome this difficulty. It also uses two criteria for the survival selection, but replaces the second criterion of the crowding distance in the NSGA-II by a reference point mechanism. Other components are the same as in the NSGA-II, see Algorithm 5.

We now give a brief introduction to the reference point mechanism. After dividing the combined population $R_t$ into serval fronts, all fronts $F_i$ with $i < i^*$ are selected and denoted as $Z_t$. Following the first theory work for the NSGA-III [WD23], we use the improved and more detailed normalization in [BDR19]. That is, all individuals in $Z_t$ are normalized by $f_j^n(x) = \frac{f_j(x) - \hat{z}_j^*}{\hat{z}_j^{\mathrm{nad}} - \hat{z}_j^*}$, where $\hat{z}_j^*$ and $\hat{z}_j^{\mathrm{nad}}$ are the ideal point estimate and the Nadir point estimate of objective $j$. Each normalized individual is then associated with a reference point with the smallest distance. Finally it repeatedly selects the reference point with the fewest already-chosen solutions (breaking ties randomly), then adds the unselected solution closest to that reference point (again breaking ties randomly) until $N - \sum_{i=1}^{i^*-1} |F_i|$ number of solutions are selected. See Algorithm 6 for more details.

The runtime of the NSGA-III is studied via the theoretical means since 2023, see [WD23, ODNS24, WD24]. Those works all focused on the performance to cover the full Pareto front. Very recently, Deng et al. [DZD25] established the first approximation guarantee of the NSGA-III and proved that the number of reference points is more important than the population size, which appeared to be an important parameter for the NSGA-II [ZD22, DQ23b]. Until now, there is no other approximation

**Algorithm 4:** Normalization with threshold parameter $\epsilon_{\text{nad}}$ [BDR19]

---

**Input:** $F_1, \ldots, F_{i^*}$ : non-dominated fronts; $f = (f_1, \ldots, f_m)$: objective function;
$z_j^w \in \mathbb{R}^m$ : observed max in each objective; $z_j^* \in \mathbb{R}^m$ : observed min in each objective;
$E \subseteq \mathbb{R}^m$ : extreme points of previous iteration, initially $\{\infty\}$;

**1 for** $j = 1, 2, \ldots, m$ **do**

**2**     $\hat{z}_j^* = \min\{z_j^*, \min_{z \in R_t} f_j(z)\}$;

**3**     $z_j^w = \max\{z_j^w, \max_{z \in R_t} f_j(z)\}$;

**4**     Determine an extreme point $e^{(j)}$ in the j-th objective from $R \cup E$ using an achievement scalarization function;

**5**     $E = E \cup \{e^{(j)}\}$;

**6** $valid$ = False;

**7 if** $e^{(1)}, \ldots, e^{(m)}$ are linearly independent **then**

**8**     $valid$ = True;

**9**     Let $H$ be the hyperplane spanned by $e^{(1)}, \ldots, e^{(m)}$;

**10**     **for** $j = 1, 2, \ldots, M$ **do**

**11**        Determine the intercept $I_j$ of $H$ with the $j$-th objective axis;

**12**        **if** $I_j \geq \epsilon_{\text{nad}}$ and $I_j \leq z_j^w$ **then**

**13**           $\hat{z}_j^{\text{nad}} = I_j$;

**14**        **else**

**15**           $valid$ = False;

**16**           **break**;

**17 if** $valid$ = False **then**

**18**     **for** $j = 1, \ldots, M$ **do**

**19**        $\hat{z}_j^{\text{nad}} = \max_{x \in F_1} f_j(x)$;

**20 for** $j = 1, 2, \ldots, m$ **do**

**21**     **if** $\hat{z}_j^{\text{nad}} < \hat{z}_j^* + \epsilon_{\text{nad}}$ **then**

**22**        $\hat{z}_j^{\text{nad}} = \max_{x \in F_1 \cup \cdots \cup F_{i^*}} f_j(x)$;

**23** Define $f_j^n(x) = \frac{f_j(x) - \hat{z}_j^{\min}}{\hat{z}_j^{\text{nad}} - \hat{z}_j^{\min}}$ for each $x \in \{0, 1\}^n$ and $j = 1, \ldots, m$;

---

theory for the NSGA-III. Before we prove that the NSGA-III satisfies Property $\mathcal{A}$, we first show the following lemma that the extremal objective values in the combined population $R_t$ will pass on the next population $P_{t+1}$. Note that Deng et al. [DZD25] proved the optimal setting of $N = N_r$ for approximating ONEMINMAX, and note that Deb and Jain [DJ14] suggests $N \approx N_r$ for the general setting. Here we only consider the setting of $N = N_r$. It is not difficult to see from the proofs that our results also hold for $N \geq N_r$.

---

**Algorithm 5:** NSGA-III [DJ14]

---

**1** Generate $P_0$ by selecting $N$ solutions uniformly and randomly from $\{0, 1\}^n$ with replacement;

**2 for** $t = 0, 1, 2, \ldots$ **do**

**3**     Generate the offspring population $Q_t$ with size $N$;

**4**     Use fast-non-dominated-sort() [DPAM02] to divide $R_t = P_t \cup Q_t$ into fronts $F_1, F_2, \ldots$;

**5**     Find $i^* \geq 1$ such that $|\bigcup_{i=1}^{i^*-1} F_i| < N$ and $|\bigcup_{i=1}^{i^*} F_i| \geq N$;

**6**     $Z_t = \bigcup_{i=1}^{i^*-1} F_i$;

**7**     Use Algorithm 6 to select $\tilde{F}_{i^*} \subseteq F_{i^*}$ such that $|Z_t \cup \tilde{F}_{i^*}| = N$;

**8**     $P_{t+1} = Z_t \cup \tilde{F}_{i^*}$;

---

**Lemma 10.** *Let $N = N_r \geq 2n + 3$ and $\epsilon_{nad} \geq n\varepsilon$. Consider using the NSGA-III to optimize* $\text{LF}'_\varepsilon$ *with problem size $n$. Define $z_j^{\min} := \min\{f_j(x) \mid x \in R_t\}$ and $z_j^{\max} := \max\{f_j(x) \mid x \in R_t\}$, $j = 1, 2$.*

**Algorithm 6:** Selection based on a set $U$ of reference points when maximizing $f$ [DJ14]

**Input:** $Z_t$: the multi-set of already selected individuals;

  $F_t^{i^*}$: the multi-set of individuals to choose from;

  $f_n$: Normalize($f$, $Z = Z_t \cup F_t^{i^*}$);

1  Associate each individual $x \in Z_t \cup F_t^{i^*}$ to the reference point rp($x$) based on the smallest distance to the reference rays;

2  For each reference point $r \in U$, initialize $\rho_r := |\{x \in Z_t \mid \text{rp}(x) = r\}|$;

3  Initialize $\tilde{F}_t^{i^*} = \emptyset$ and $U' = U$;

4  **while** True **do**

5      Let $r_{\min} \in U'$ such that $\rho_{r_{\min}}$ is minimal (breaking ties randomly);

6      Let $x_{r_{\min}} \in F_t^{i^*} \setminus \tilde{F}_t^{i^*}$ which is associated with $r_{\min}$ and minimizes the distance between $f_n(x_{r_{\min}})$ and $r_{\min}$ (breaking ties randomly);

7      **if** $x_{r_{\min}}$ exists **then**

8         $\tilde{F}_t^{i^*} = \tilde{F}_t^{i^*} \cup \{x_{r_{\min}}\}$;

9         $\rho_{r_{\min}} = \rho_{r_{\min}} + 1$;

10        **if** $|Z_t| + |\tilde{F}_t^{i^*}| = N$ **then**

11           **return** $\tilde{F}_t^{i^*}$

12        **else**

13           $U' = U' \setminus \{r_{\min}\}$

*Then the next parent population $P_{t+1}$ will preserve two individuals $x, y$ such that $f_1(x) = z_1^{\min}$ and $f_1(y) = z_1^{\max}$.*

With Lemma 10, we easily see that once $(0, n\varepsilon)$ and $(n\varepsilon, 0)$ are covered by $R_t$, they will be covered by the next population. The following lemma shows that after $(0, n\varepsilon)$ and $(n\varepsilon, 0)$ are covered by $R_t$, Property $\mathcal{A}$ will be satisfied.

**Lemma 11.** *Let $N = N_r \geq 2n + 3$ and $\epsilon_{nad} \geq n\varepsilon$. Consider using the NSGA-III to optimize $\text{LF}'_\varepsilon$ with problem size $n$. Assume that at some iteration $t$, the two extreme points $(0, n\varepsilon)$ and $(n\varepsilon, 0)$ are covered by the combined parent and offspring population $R_t = P_t \cup Q_t$. If $R_t$ contains an individual $x$ with $|x'|_1 = k$, then the next parent population $P_{t+1}$ also contains an individual $y$ with $|y'|_1 = k$, and covers $(0, n\varepsilon)$ and $(n\varepsilon, 0)$ as well.*

With Lemma 10, it is not difficult to obtain the runtime to cover $(0, n\varepsilon)$ and $(n\varepsilon, 0)$. Then from Lemma 11 asserting that Property $\mathcal{A}$ is satisfied, we use the general approximation theorem (Theorem 7) to obtain an $O(n^2 \log n)$ (when setting $N = \Theta(n)$) expected runtime to reach an additive $\varepsilon$-approximation w.r.t. $\text{LF}'_\varepsilon$.

**Theorem 12.** *Let $N = N_r \geq 2n + 3$, $\epsilon_{nad} \geq n\varepsilon$ and $p_c \in [0, 1)$. Consider using the NSGA-III with uniform selection and one-bit mutation or bit-wise mutation to optimize $\text{LF}'_\varepsilon$ with problem size $n$. Then after an expected number of $O(Nn \log n)$ fitness evaluations, the population is an additive $\varepsilon$-approximation of $\text{LF}'_\varepsilon$.*

### 4.3 SMS-EMOA

The SMS-EMOA [BNE07] can be seen as a steady-state variant of the NSGA-II in which crowding distance is replaced by the hypervolume contribution indicator. In each generation, it generates one offspring and then only removes one individual from $R_t$. The hypervolume indicator is the most widely used measure for approximation quality in evolutionary multi-objective optimizations [SIHP20]. Given a reference point $r$, the hypervolume of a population $S$ is calculated as

$$\text{HV}_r(S) = \mathcal{L}\left( \bigcup_{u \in S} \{h \in \mathbb{R}^m \mid r \leq h \leq f(u)\} \right),$$

where $\mathcal{L}$ is the Lebesgue measure. The hypervolume contribution of an individual $x \in F_{i^*}$ is defined as $\Delta_r(x, F_{i^*}) := \text{HV}_r(F_{i^*}) - \text{HV}_r(F_{i^*} \setminus \{x\})$ for $x \in F_{i^*}$. Algorithm 7 gives the pseudo code of

**Algorithm 7:** SMS-EMOA [BNE07]

1 Generate $P_0$ by selecting $N$ solutions uniformly and randomly from $\{0,1\}^n$ with replacement;
2 **for** $t = 0, 1, 2, \ldots,$ **do**
3      Generate one offspring $y$;
4      Use fast-non-dominated-sort() [DPAM02] to divide $R_t = P_t \cup \{y\}$ into $F_1, \ldots, F_{i^*}$ ;
5      Calculate $\Delta_r(z, F_{i^*})$ for all $z \in F_{i^*}$ and find $D = \arg\min_{z \in F_{i^*}} \Delta_r(z, F_{i^*})$;
6      Uniformly at random pick $q \in D$ and $P_{t+1} = R_t \setminus \{q\}$ ;

---

the SMS-EMOA. It fits into our general MOEA framework (Algorithm 1) with fixed population size of $N$, offspring population size $\lambda = 1$, and the survival selection based on hypervolume contribution.

Although Bian et al. [BZLQ23] and Zheng and Doerr [ZD24a] have analyzed the runtime of the SMS-EMOA on bi- and many-objective benchmarks, its theoretical approximation performance remains unstudied. Brockhoff et al. [BFN08] proved that the $(\mu + 1)$-SIBEA algorithm, a simplified version of the SMS-EMOA without fast non-dominated sorting, achieves a multiplicative $\varepsilon$-approximation of the LARGEFRONT$_\varepsilon$ variant $LF_\varepsilon$ in expected $O(\mu n \log n)$ number of fitness evaluations. No approximation results for the SMS-EMOA or $(\mu + 1)$-SIBEA on LF$'_\varepsilon$ are given. As in the previous sections (also similar to the proof of $(\mu + 1)$-SIBEA on $LF_\varepsilon$ [BFN08]), we first show that the SMS-EMOA has Property $\mathcal{A}$ w.r.t. LF$'_\varepsilon$.

**Lemma 13.** *Let $N \geq \frac{n}{2} + 1$ and $r = (r_1, r_2)$ with $r_1 \leq -\varepsilon$, $r_2 \leq -\varepsilon$. Consider using the SMS-EMOA to optimize LF$'_\varepsilon$ with problem size $n$. Assume that at some iteration $t$ the combined parent and offspring population $R_t$ contains an individual $x$ with $|x'|_1 = k$. Then the next parent population $P_{t+1}$ contains an individual $y$ such that $|y'|_1 = k$.*

Combining Lemma 13 with our general theorem (Theorem 7), we obtain the expected runtime of $O(Nn \log n)$, which is $O(n^2 \log n)$ when $N = \Theta(n)$, required to reach an additive $\varepsilon$-approximation w.r.t. LF$'_\varepsilon$.

**Theorem 14.** *Let $N \geq n/2 + 1$, $r = (r_1, r_2)$ with $r_1 \leq -\varepsilon$, $r_2 \leq -\varepsilon$ and $p_c \in [0, 1)$. Consider using the SMS-EMOA to optimize LF$'_\varepsilon$ using uniform selection and one-bit mutation or bit-wise mutation with problem size $n$. Then after an expected number of $O(Nn \log n)$ fitness evaluations, the population is an additive $\varepsilon$-approximation w.r.t. LF$'_\varepsilon$.*

### 4.4 SPEA2

The SPEA2 algorithm [ZLT01] is one of the most popular MOEAs. In the survival selection at generation $t$, it creates a new parent population $P_{t+1}$ by selecting all non-dominated solutions from $R_t$. If $|P_{t+1}|$ is smaller than the population size $N$, then it is supplemented with the best dominated individuals, determined by the strength and density estimation. If the number of non-dominated individuals exceeds the population size $N$, a truncation operator is used to iteratively remove solutions from $P_{t+1}$ until $|P_{t+1}| = N$. Let $\sigma_u^k$ denote the Euclidean distance (in the objective space) of the individual $u$ to its $k$-th nearest neighbor in $R_t$ with $k = \sqrt{N + \lambda}$. At each removal, an individual $u$ is chosen for removal with $u \leq_d v$ for all $v \in P_{t+1}$, where

$$u \leq_d v :\Leftrightarrow \forall 0 < k < |P_{t+1}| : \sigma_u^k = \sigma_v^k \vee$$
$$\exists 0 < k < |P_{t+1}| : [(\forall 0 < l < k : \sigma_u^l = \sigma_v^l) \wedge \sigma_u^k < \sigma_v^k].$$

In other words, at each removal, it removes the individual with the smallest nearest-neighbor distance and ties are broken by comparing their second-nearest distances and so forth. Once a solution is removed, its distances to other solutions are no longer considered. See Algorithm 8 for more details. The SPEA2 fits into the general MOEA framework (Algorithm 1) with uniform parent selection, and the survival selection based on the truncation operator.

The first runtime analysis of the SPEA2 was conducted very recently [RBLQ24] and proved the runtime bounds for the SPEA2 on three commonly used multi-objective problems, namely $m$ONEMINMAX, $m$LOTZ, and mOJZJ. Prior work by Horoba and Neumann [HN09] studied the approximation performance of RADEMO (a simplified version of the SPEA2), and proved that it achieves an additive $\varepsilon$-approximation w.r.t. LF$'_\varepsilon$ in polynomial runtime. Till now, no approximation

**Algorithm 8:** SPEA2[[ZLT01]]

---

1   $Q_0 \leftarrow \lambda$ solutions uniformly and randomly selected from $\{0,1\}^n$ with replacement and $P_0 \leftarrow \emptyset$;
2   **for** $t = 0, 1, 2, \ldots$ **do**
3      $P_{t+1} \leftarrow$ non-dominated solutions in $R_t = P_t \cup Q_t$;
4      **if** $|P_{t+1}| > N$ **then**
5         Reduce $P_{t+1}$ by means of the truncation operator;
6      **else if** $|P_{t+1}| < N$ **then**
7         Fill $P_{t+1}$ with dominated individuals in $R_t$;
8      **for** $i = 0, 1, 2, \ldots, \lambda$ **do**
9         Generate one offspring $x'$;
10        $Q_{t+1} \leftarrow Q_{t+1} \cup \{x'\}$;

---

guarantees for the SPEA2 on $\mathrm{LF}'_\varepsilon$ were given. As in the previous sections, we first prove that the SPEA2 maintains Property $\mathcal{A}$ required for our general approximation theorem.

**Lemma 15.** *Let $N \geq n/2 + 2$. Consider using the SPEA2 to optimize $\mathrm{LF}'_\varepsilon$ with problem size $n$. If at some iteration, the combined population $R_t$ contains an individual $x$ with $|x'|_1 = k$, then the next population $P_{t+1}$ will also include an individual $y$ with $|y'|_1 = k$.*

With Lemma 15, we derive an expected runtime of $O(n^2 \log n)$ to reach an additive $\varepsilon$-approximation w.r.t. $\mathrm{LF}'_\varepsilon$, by setting $N = \Theta(n)$ in our general approximation theorem.

**Theorem 16.** *Let $N \geq n/2 + 2$ and $p_c \in [0, 1)$. Consider using the SPEA2 with uniform selection and one-bit mutation or bit-wise mutation to optimize $\mathrm{LF}'_\varepsilon$ with problem size $n$. Then after an expected $O(Nn \log n)$ number of fitness evaluations, the population is an additive $\varepsilon$-approximation w.r.t. $\mathrm{LF}'_\varepsilon$.*

## 5   Conclusion and Discussion

The question of why popular MOEAs are popular in practice was not yet convincingly answered by theoretical methods. The few results indicating advantages only considered the performance to cover the full Pareto front on newly designed benchmarks. This work tackled this question by considering the approximation ability of several popular MOEAs on the established LARGEFRONT$'_\varepsilon$ benchmark. In contrast to the $2^{\Omega(n^{1/4})}$ fitness evaluations (with high probability) the GSEMO takes to reach an additive $\varepsilon$-approximation, we gave a general theorem showing polynomial runtimes for any MOEA with Property $\mathcal{A}$, and proved $O(n^2 \log n)$ expected runtimes for four widely used MOEAs, namely NSGA-II, NSGA-III, SMS-EMOA, and SPEA2. The reason for this advantage is the second selection criterion of these popular MOEAs ensuring a good diversity in the survival selection, compared to the GSEMO that relies only on the dominance criterion. This is the first mathematical runtime analysis showing and explaining the superiority of popular MOEAs over simpler ones like the GSEMO for the central task of computing good approximations to the Pareto front. This work also is the first approximation study for the SMS-EMOA.

Our results and proofs suggest advantages in approximation also for other benchmarks with large number of Pareto front points, but a more thorough and rigorous analysis on more general benchmark classes is necessary to support this claim.

## Acknowledgments and Disclosure of Funding

This work was supported by National Natural Science Foundation of China (Grant No. 62306086, 62350710797), Guangdong Basic and Applied Basic Research Foundation (Grant No. 2025A1515011936), Xinjiang Tianshan Innovative Research Team (2025D14009), and Science, Technology and Innovation Commission of Shenzhen Municipality (Grant No. GXWD20220818191018001). This research benefited from the support of the FMJH Program PGMO.

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
