# OpenReview forum: "Why Popular MOEAs Are Popular: Proven Advantages in Approximating the Pareto Front"
_NeurIPS.cc/2025/Conference — NeurIPS 2025 poster_

### Official Review · Reviewer_2K1g · 2025-06-02

**Clarity:** 3
**Significance:** 2
**Originality:** 3
**Rating:** 4
**Confidence:** 4

**Summary:**

This paper analyzes popular Multi-Objective Evolutionary Algorithms (MOEAs) to explain their practical advantages in approximating the Pareto front. It addresses shortcomings in previous analyses, which often focused on computing the full Pareto front. The study analyzes how existing popular MOEAs approximate the Pareto front of the LARGEFRONT benchmark, specifically LF\epsilon. The paper proves that popular MOEAs like NSGA-II, NSGA-III, SMS-EMOA, and SPEA2 require an expected O(n^2logn) function evaluations to compute an additive \epsilon-approximation of the Pareto front of the LF\epsilon benchmark. This is a significant improvement over the simple GSEMO algorithm, which has previously been proven to take an exponential runtime to approximate the same benchmark.

**Questions:**

1. The paper uses the additive \epsilon-approximation definition. Do the authors know if the conclusions of the paper still holds if other definitions of approximation is used? In fact, why did the authors choose the additive version of \epsilon-dominance instead of (1+\epsilon)u \geq v?
2. Can the authors further elaborate on why LF\epsilon is chosen, and if it can represent a larger class of benchmarks?
3. The authors very briefly mention in the introduction that the advantage of MOEAs over GSEMO is the fact that they maintain a diverse population. However, the proofs of how the diversity mechanisms ensure property A and thus ensure the approximation guarantee are all pushed into the supplementary. It would be nice to see an illustration of this in the main paper, or even a dedicated section exploring the different diversity mechanisms, to deepen the analysis provided by the authors.

**Ethical Concerns:**

["NO or VERY MINOR ethics concerns only"]

**Final Justification:**

The paper provides insights on several popular MOEAs. As other reviewers have mentioned, the paper is strong is its core contribution and the problem studied here is valuable. This is mainly the reason that I think the paper should be accepted. The paper has its drawbacks. For example, the lack of empirical studies is the main reason why I did not raise my score more.

**Limitations:**

yes

**Quality:**

2

**Strengths And Weaknesses:**

Strengths:
1. The paper provides a well-founded mathematical analysis of popular MOEAs for the task of approximating the Pareto front of the LF\epsilon benchmark. The focus on approximation instead of recovering the full Pareto frontier is relevant since in practice, the size of the frontier could be exponential and impossible to be recovered fully.
2. The paper covers a wide range of MOEAs, like NSGA-II, NSGA-III, SMS-EMOA, and SPEA2. This broad coverage enhances the impact and relevance of the findings.
3. The paper uses the established benchmark LF\epsilon, which contrasts with previous works that might rely on specialized newly designed benchmarks.
4. The introduction of the general approximation theory in theorem 7 provides a venue for the analysis of more EAs under the same framework.
5. The paper is well-written, clearly organized, and easy to read.

Weaknesses:
1. A major weakness is the lack of empirical validation. Even though the paper is theoretical in nature, its practicality could be greatly improved by including experiments that complement the theoretical results.
2. The paper uses only one benchmark LF\epsilon. It brings into question on the generalizability of the results on other benchmarks.

---

> ### Author Rebuttal · Authors · 2025-07-31
>
> Thanks a lot for your comments on weaknesses and questions. In the following, we respond to them one by one. With the following responses, we hope that you are convinced to raise your grade.
> - **(W1) Major weakness: no empirical validation.** The reviewer agrees that this is a pure theory paper. (1) Indeed, major AI conferences, like NeurIPS, AAAI, IJCAI, etc., accept the pure theory papers, not only for our evolutionary theory community (e.g.[1,2,3]), but also for other community (e.g. [4,5,6]). (2) Also mentioned by the second reviewer, empirical illustrations would be good. However, he/she also mentioned this is just a limitation, not the weakness, and "do not undermine the core contribution". (3) Limited space. The reviewer also noticed (in his/her Q3) that all our proofs are pushed into the supplementary material due to the limited space. We do not have more space for experiments as we set it the priority to sufficiently demonstrate our theoretical contribution. Hence, we kindly ask the reviewer to view it as a limitation not a major weakness for a negative score.
> - **(W2 \& Q2) Only one LargeFront benchmark.** (1) We first note that LargeFront is not one benchmark but a class of benchmarks w.r.t. $\varepsilon$ (See Definition 1, Page 2). (2) We would like to point its relation to the continuous problems. As stated in the second paragraph in Section 2.1, LargeFront has "exponential Pareto front points", and is much closer to "the continuous optimization as a segment of a continuous curve in the Pareto front contains infinite points". Our results on LargeFront might shed some light on the understanding of the popular MOEAs for real-world continuous optimization.
> - **(Q1) Why additive $\varepsilon$-approximation?** We chose the additive $\varepsilon$-approximation because it aligns naturally with the structure of the $LF_\varepsilon'$ benchmark (see [HN09] also used additive $\varepsilon$-approximation for the $LF_\varepsilon'$ benchmark), and we chose $LF_\varepsilon'$ as it "shows more similarity to the arguably most popular ONEMAX benchmark" (See the second paragraph in Section 2.1). For the question of whether the conclusion still holds for other approximation measures on the $LF_\varepsilon'$ benchmark class, we want to say that we haven't considered it yet. We note that for the technical part to establish our results, we essentially try to calculate the time for the population to cover all $k$ ones for $k=0,1,\dots,n/2$ (that is, the time to witness a population $P$ with $P\cap ${$ x\mid |x'|_1=k $} $\neq \emptyset$ for all $k\in[0..n/2]$) (See Lemma 4). Hence, from this point of view, we could transfer our results to other measures by considering how well the population with the above characteristic approximates, but we are not sure how tight this calculation is. Definitely, it will be an interesting work to do in our future.
> - **(Q3) Omitted proofs in the main text.** Thanks a lot for your nice suggestions for adding more illustrations for the omitted proofs and findings. We will try to find some space for it in our future version.
>
>
> ## References
>
> [1] Per kristian Lehre, Shishen Lin. No Free Lunch Theorem and  Black-Box Complexity Analysis for  Adversarial Optimisation. In Proceedings of the 38th Annual Conference on Neural Information Processing Systems, NeurIPS 2024, pages 121570-121597.
>
> [2] Sacha Cerf, Benjamin Doerr, Benjamin Hebras, Yakob Kahane, and Simon Wietheger. The First Proven Performance Guarantees for the Non-Dominated Sorting Genetic Algorithm II (NSGA-II) on a Combinatorial Optimization Problem. In International Joint Conference on Artificial Intelligence, IJCAI 2023, pages 5522-5530.
>
> [3] Duc-Cuong Dang, Andre Opris, Bahare Salehi, and Dirk Sudholt. A proof that using crossover can guarantee exponential speed-ups in evolutionary multi-objective optimisation. In Conference on Artificial Intelligence, AAAI 2023, pages 12390–12398.
>
> [4] Holden Lee, Jianfeng Lu, and Yixin Tan. Convergence for score-based generative modeling with polynomial complexity. In Proceedings of the 36th Annual Conference on Neural Information Processing Systems, NeurIPS 2022, pages 22870-22882.
>
> [5] Cornelius Brand, Robert Ganian, and Kirill Simonov. A Parameterized Theory of PAC Learning. In Conference on Artificial Intelligence, AAAI 2023, pages 6834–6841.
>
> [6] Christel Baier, Florian Funke, and Rupak Majumdar. A game-theoretic account of responsibility allocation. In International Joint Conference on Artificial Intelligence, IJCAI 2021, pages 1773-1779.

---

> > ### Comment · Reviewer_2K1g · 2025-08-05
> >
> > Thanks the authors for addressing my questions and concerns. I do still feel like empirical results could greatly strengthen the paper. As for the authors' argument on lack of space, I actually find the use of space in this paper to be rather wasteful. For example, there are blocks with code format to illustrate every algorithm studied. I find this to be unnecessary. As the algorithms studied here are generally well known in the EA community, and the original papers proposing these algorithms are easily accessible, details on the algorithm could actually go in the appendix for readers who need to look them up and a paragraph to describe each algorithm in the main text would suffice. However, I have decided to raise my score since the authors have addressed my other concerns.

---

> > > ### Author Response · Authors · 2025-08-05
> > >
> > > Thanks again for your comments. The reason we gave the pseudocode is that for a mathematical analysis, it is really important that the precise code is given, and small variations can make a huge difference. Unfortunately, many of the works originally proposing the algorithms are not that precise. That said, we will carefully check where the presentation can be condensed as you suggest. Thanks for your time and care!

---

### Official Review · Reviewer_RTSU · 2025-06-19

**Clarity:** 3
**Significance:** 3
**Originality:** 3
**Rating:** 5
**Confidence:** 4

**Summary:**

This paper aims to address the gap in understanding why popular MOEAs outperform simpler MOEAs (Popular MOEAs have more sophisticated environmental selection strategies). The theoretical properties of four popular MOEAs are revealed using established benchmark problems.

**Questions:**

- Why is MOEA/D excluded from the analysis? To my knowledge, it is a representative decomposition-based MOEA.
- What is the rationale for selecting SPEA2? I think that analyzing either NSGA-II or SPEA2 would be sufficient.
- What is the significance of the exponential growth in the number of Pareto-optimal objective vectors in LARGEFRONT?
- The mating pool selection is random in all analyzed MOEAs. Would it be possible to consider the mating pool selection strategy in the original MOEAs?

**Ethical Concerns:**

["NO or VERY MINOR ethics concerns only"]

**Final Justification:**

After careful consideration of all the discussions, I have decided to increase my score to support the acceptance of this work.

My major concern is the consistency of the assumptions in the theorems. The author(s) have agreed to provide proofs based on consistent assumptions in the revised version.

**Limitations:**

The conclusion section provides a brief discussion.

**Quality:**

3

**Strengths And Weaknesses:**

Strengths:
- The paper is clearly and logically structured.
- It addresses a research gap by providing a theoretical analysis of existing representative problems.
- The analysis is conducted under a more appropriate optimality criterion.
- A generalized theorem (Theorem 7) is proposed.

Weaknesses:
- The term "sequential version" introduced in the abstract is not referenced later; "steady-state" is instead used. It is recommended to maintain consistent terminology throughout the paper.
- In the introduction, the paragraph beginning with "However, ..." is difficult to follow despite being a continuation of the previous paragraph. It is suggested to provide specific references and examples in this paragraph directly to clarify the points.
- The additive $\epsilon$-approximation seems to be applicable only to LARGEFRONT.
- The paper does not specify the crossover operator employed. Additionally, there is no explanation provided for setting the crossover probability $p_c=0$ in SMS-EMOA and SPEA2.

---

> ### Author Rebuttal · Authors · 2025-07-31
>
> Thanks a lot for your comments and questions. We believe that our following responses will properly address them (we also note that some of the concerns might come from a misunderstanding), and hope that it raises the appreciation.
> - **(W1) Writing suggestion: Inconsistent terminology for "sequential version" and "steady-state".** Indeed, "sequential version" and "steady-state" are two strategies. We guess the reviewer means the inconsistent terminology for "sequential version" and "using the current crowding distance". Thanks, and we will modify our writing in our future version.
> - **(W2) Writing suggestion: One paragraph in the introduction.** Thanks, and we will reform it in our future version.
> - **(W3) The additive $\varepsilon$-approximation only applicable to LargeFront.** It is not ture. As stated in Section 2.2 (Page 3), this additive $\varepsilon$-approximation was "first defined in [LTDZ02], that relaxes the usual dominance relation by allowing an additive slack $\varepsilon$ in each objective". Also from the formal definition (See Definition 3), no restrictions on problem are required. Also note that LargeFront was proposed in 2009, but the additive $\varepsilon$-approximation was proposed in 2002. In summary, the additive $\varepsilon$-approximation is a general measure and applicable to other problems. We will make this point clearer in the future version.
> - **(W4) Crossover operator.** Thanks for pointing this out. We noticed that a similar question is raised by the first reviewer. (1) Indeed, our general approximation theorem (Theorem 7) states "Let the crossover rate $p_c\in[0,1)$", which means this result holds no matter the crossover is used (when $p_c\in(0,1)$) or not (when $p_c=0$). We admit that we are missing the detail about the crossover, and this work uses the single-point crossover as used in the original work of the NSGA-II for binary code. (2) The reason to choose the crossover rate of 0 for the SMS-EMOA and SPEA2: we just follow the same setting (that only mutation is used) in previous theoretical analysis for these two algorithms, see [BZLQ23,ZD24c,RBLQ24]. We will make them clearer in our future version.
> - **(Q1) Excluded MOEA/D.** Thanks for this nice question, and we note that this is also raised by the first reviewer. The reason comes from two parts. (1) The existing theoretical results did not discuss the performance of the MOEA/D, when they tried to understand why popular MOEAs are popular (see the second paragraph in Section 1). (2) We expected a clean and clear understanding on the effect of algorithm component. The analyzed popular MOEAs share the similarity that they do not decompose the multi-objective problems and consider one more criterion in the survival selection, compared with the simple GSEMO where only dominance criterion is used. However, MOEA/D is a different framework. But we thank the reviewer for pointing this out and we will analyze the approximation of the MOEA/D as our interesting future work.
> - **(Q2) Why SPEA2 analyzed? Either NSGA-II or SPEA2 sufficient?**  We both agree that SPEA2 is one of the popular MOEAs. We note that NSGA-II and SPEA2 can behave differently from our theoretical perspective. For example, NSGA-II is proven to be inefficient (at least exponential runtime) to cover the full Pareto front for the OMM-type problems when the number of objectives is at least three (see [ZD24b]). However, [RBLQ24] showed the polynomial runtime for the above problems. Hence, we also analyzed the SPEA2 in addition to the NSGA-II.
> - **(Q3) Significance of the exponential Pareto front points.** Good question. Indeed, as stated in the second paragraph in Section 2.1, "the theory of evolutionary algorithms is majorly built on the discrete space'' and other widely analyzed benchmarks "contain the polynomial number of Pareto (front) points". LargeFront has "exponential Pareto front points", and is much closer to "the continuous optimization as a segment of a continuous curve in the Pareto front contains infinite points". Our results on LargeFront might shed some light on the understanding of the popular MOEAs for real-world continuous optimization.
> - **(Q4) Other mating pool selection.** Yes, it is possible to consider other mating pool selection, but case-by-case probabilistic calculation is needed. A good evidence is that in the runtime analysis of the NSGA-II [ZD23], random selection, fair selection, and binary tournament selection are analyzed, and all lead to the same runtime complexity. We conjecture that our results also hold for other mating pool selections.

---

> ### Comment · Reviewer_RTSU · 2025-08-01
>
> ### About W3
> The reason I introduced W3 is that I noticed both Sections 2.1 and 2.2 use the symbol $\varepsilon$. The authors' response seems to suggest that some existing handcrafted problems admit an analytical additive $\varepsilon$-approximation. For a handcrafted problem, if it admits an analytical additive $\varepsilon$-approximation, does this require a specialized formulation of the problem, such as incorporating $\varepsilon$ explicitly in the formulation?
>
> Furthermore, I think it might be clearer to first present the fundamental definitions and the concept of additive $\varepsilon$-approximation, followed by the introduction of LARGEFRONT and then Theorems 4 & 5.
>
>
> ### About W4 & Q1
> The response has not addressed W4, which I consider crucial for the coherence and completeness of the work.
>
> I do not understand the claim that "they do not decompose the multi-objective problems and consider one more criterion in the survival selection." By this logic, one could argue that "NSGA-II, NSGA-III, SPEA2, and MOEA/D do not use scalar indicators for environmental selection, and thus SMS-EMOA should not be considered."
>
> The responses to W4 and Q1 have reinforced my view that this work is incremental, simply extending previous results without providing a significant breakthrough.
>
>
> ### About Q3
> Although the size of the Pareto front grows exponentially, I have observed that the number of elements in the additive $\varepsilon$-approximation increases linearly with $n$. Does this imply that the analysis simplifies to a Pareto front with linear growth?

---

> > ### Author Response · Authors · 2025-08-02
> > **W4**
> >
> > Hi, could you specify please in what sense our answer marked W4 has not addressed your concern W4? We thought that we had completely discussed the points you raised. Thanks a lot!

---

> > > ### Comment · Reviewer_RTSU · 2025-08-03
> > >
> > > I think it is important to maintain consistent settings when analyzing different algorithms.

---

> > > > ### Author Response · Authors · 2025-08-03
> > > > **W4**
> > > >
> > > > - **Further Comments about W4.** Okay, now we guess we understand what you mean. Do you mean that you want a fair comparison by using consistent settings for different algorithms? That is, we should not compare the NSGA-II and NSGA-III with crossover to the SMS-EMOA and SPEA2 without crossover. We agree with it, and we do compare different algorithms in the consistent setting in our submission. We mentioned $p_c=0$ for the SMS-EMOA and SPEA2 because the existing theoretical works used this setting. Indeed, if the SMS-EMOA and SPEA2 are equipped with the single-point crossover with $p_c\in[0,1)$, our results also hold. Technically, as we mentioned, our general approximation theorem (Theorem 7) states "Let the crossover rate $p_c\in[0,1)$", which means the result holds no matter the crossover is used (when $p_c\in(0,1)$) or not (when $p_c=0$). The reason for that the crossover does not affect the runtime bound is that, when calculating the probability of generating a desired solution (see Line 471 in the supplementary material), we consider the term $(1-p_c)$, that is, we focus on the probability of generating the desired solution via mutation only (i.e., when crossover is not activated). We guess now we somehow realize why $p_c=0$ for the SMS-EMOA and SPEA2 caused your confusion, and will make it clearer that our results hold for the above algorithms using the single-point crossover with $p_c\in[0,1)$ in our future version. We hope that we have addressed your concern now. If not or if we misunderstand your point on "consistent settings", we are happy to have a further discussion.

---

> > > > > ### Comment · Reviewer_RTSU · 2025-08-04
> > > > >
> > > > > Thank you for your response.
> > > > >
> > > > > ### About W4
> > > > > I reviewed the paper again, and I observed that Theorems 9, 12, 14, and 16 in Section 4 analyze only the case of mutation without crossover. However, Section 3 adopts different assumptions. This discrepancy may confuse audiences as to why the analysis in Section 4 does not follow the assumptions presented in Section 3.
> > > > >
> > > > > Furthermore, the notation for population size differs between Theorems 9 and 12 and Theorems 14 and 16, and I fail to understand the rationale behind this inconsistency.
> > > > >
> > > > > ### About Q1
> > > > > I still don't understand why MOEA/D is not analyzed. MOEA/D fundamentally does not solve single-objective subproblems separately but rather employs a collaborative scheme. Its theoretical analysis should differ significantly from that of single-objective EAs and may be representative in the context of multi-objective optimization. Is this omission due to the complexity of its analysis?
> > > > >
> > > > > ### About Q3
> > > > > Based on your response, I think the novelty of Theorems 9, 12, 14, and 16 is limited. They may easily be derived using methodologies established in previous works. Consequently, the main contribution of this paper seems to be Theorem 7.

---

> > > > > > ### Author Response · Authors · 2025-08-04
> > > > > > **On Q1:**
> > > > > >
> > > > > > We agree that it would be nice to have also the MOEA/D included in this work. For two reasons we did not do so.
> > > > > >
> > > > > > First, the MOEA/D is very different from the domination-based algorithms regarded in this work. This might be not a problem for an experimental work, but it is for a mathematical analysis. We note that there is no previous work conducting both a runtime analysis of the MOEA/D and the domination-based algorithm discussed in this work. In fact, most previous works only regard a single algorithm. In this respect, we feel that our work is quite substantial and broad, and at least on par with other runtime analyses that have appeared recently at AAAI, IJCAI, and NeurIPS.
> > > > > >
> > > > > > Second, for the MOEA/D, the real difficulty is the decomposition. A decomposition that works well, e.g., for the simple LeadingOnesTrailingZeros benchmark fails for most problems obtained from monotonically transforming the objectives (even though all these problems have a small Pareto front). For that reason, the main focus of our work is less interesting for the MOEA/D.
> > > > > >
> > > > > > In summary, we agree that including the MOEA/D would have been possible, but it is less urgent and also without our paper constitutes a substantial progress. In particular, we do not think that not regarding the MOEA/D is a noteworthy weakness of our work, and a good reason to reject the paper. We could imagine that some formulations, such as "analyzing how existing popular MOEAs", led the reviewers first believe that we also regard the MOEA/D, leading to a disappointment when finally no such results were found. We are sorry for this. We shall revise our presentation accordingly, in particular, by early on specifying that we only regard domination-based algorithms.

---

> > > > > > ### Author Response · Authors · 2025-08-05
> > > > > > **About W4 & Q3**
> > > > > >
> > > > > > - **About W4.** Thanks a lot for reviewing our submission again and pointing this out. As explained in the previous responses, these results in Section 4 indeed hold for the algorithms using the single-point crossover with $p_c\in[0,1)$. We agree with you that writing the results with the setting of $p_c\in[0,1)$ in Section 4 is better for the consistency with Section 3, and we will modify them in our future version. Regarding the population size notation, we followed conventions from prior theoretical work: $\mu$ for SMS-EMOA and SPEA, and $N$ for NSGA-II and NSGA-III. Thank you for your comments on the writing. We will ensure the notation is consistent throughout in our future version.
> > > > > >
> > > > > > - **About Q3.** Thank you for your feedback. However, we respectfully disagree with your assessment that "the novelty of Theorems 9, 12, 14, and 16 is limited. They may easily be derived using methodologies established in previous works". The existing theoretical results can be divided into two categories: one for covering the full Pareto front, and the other for a good approximation of the Pareto front. For the former category, they do not involve selecting among different Pareto front points in the survival selection. In contrast, our proof of the satisfaction of Property $\mathcal{A}$ (used in Section 4) demands a much more detailed analysis, as we need to examine how additional mechanisms (e.g., crowding distance, hypervolume contribution) influence the selection among different Pareto front points. For the latter category, there is only one existing approximation result for each of the algorithms NSGA-II and NSGA-III, and all of these results are based on the OneMinMax benchmark, which is one of the simplest benchmarks in theoretical analysis. For instance, the only existing approximation guarantee for NSGA-III on OneMinMax (See [DZD25]) focuses on the impact of reference points and calculates the upper bound on the maximum empty interval size, an approach that does not directly apply to our case.

---

> > > > > > > ### Comment · Reviewer_RTSU · 2025-08-05
> > > > > > >
> > > > > > > Thank you very much to the author(s) for the patient reply.
> > > > > > >
> > > > > > > - My concern in W4 has been resolved. I sincerely hope the author(s) carefully revise or refine the proofs. I believe that establishing consistent settings will significantly improve the quality of the work. About your previous response on W4：The response attributes the issues to "a shortcoming of all previous works," which, I think, cannot be used as an excuse for not improving the paper.
> > > > > > > - The response clarifies the theoretical challenge of MOEA/D, specifically the assumption that single-objective subproblems should align with the Pareto front (is my understanding correct?). My doubts are resolved. I think this point can be included in the paper. And I think stating "early on specifying that we only regard domination-based algorithms" is not necessary.
> > > > > > > - My concern in Q3 has been resolved.
> > > > > > >
> > > > > > > After careful consideration of all the discussions, I have decided to increase my score to support the acceptance of this work. Good luck!
> > > > > > >
> > > > > > > Finally, I sincerely recommend that the author(s) consider my suggestions in the camera-ready version (if accepted) to further improve the quality of this work. I believe these improvements will greatly benefit both the audience and the EMO community.

---

> > > > > > > > ### Author Response · Authors · 2025-08-05
> > > > > > > > **Thank you!**
> > > > > > > >
> > > > > > > > Dear reviewer RTSU,
> > > > > > > >
> > > > > > > > Thank you very much for the time and care you attributed to our paper, and for accepting our arguments. This is rare these days, so we are really happy to have had you assigned to our paper. We are very proud of our work, so we will definitely follow all your suggestions to further improve it.
> > > > > > > >
> > > > > > > > Thanks and all the best
> > > > > > > > the authors of paper #1947

---

> > ### Author Response · Authors · 2025-08-03
> > **W3, Q1 & Q3**
> >
> > - **About W3.** Thank you for your further comments about W3. Regarding your question, for a handcrafted problem, the existence of an analytical additive $\varepsilon$-approximation does not necessarily require a specialized formulation of the problem like incorporating $\varepsilon$ explicitly in the formulation. In general, the quality of the approximation depends on the value of $\varepsilon$ and obviously the smaller the $\varepsilon$, the tighter the approximation, and as $\varepsilon \to 0$, the approximated set naturally approaches the true Pareto front. However, for the specific benchmark $LF'_\varepsilon$, $\varepsilon$ is explicitly embedded in the formulation of the objective function. Finally, we sincerely appreciate your suggestion regarding the organization of the content and it does help.
> >
> > - **About Q1.** Sorry for the confusion caused by our earlier statement: "they do not decompose the multi-objective problems and consider one more criterion in the survival selection". What we intended to express is that, in contrast to the simple GSEMO, which relies solely on the dominance criterion, the four popular MOEAs we studied incorporate an additional criterion in the survival selection beyond dominance and this allows for a cleaner and more transparent analysis of how different algorithmic components affect performance. As for the reason to choose the GSEMO for comparison: First, GSEMO is arguably the simplest and most theoretically analyzed MOEA and hence it serves as a natural baseline. Also the existing theoretical results usually uesd GSEMO as a reference point, when they tried to understand why popular MOEAs are popular. Second, prior work [HN09] has already shown that the GSEMO fails to achieve an additive $\varepsilon$-approximation of $LF'_\varepsilon$ in polynomial runtime (see Theorem 5), both highlighting a clear theoretical limitation and making it a meaningful reference point for understanding the impact of additional selection criteria used in advanced MOEAs.
> >
> > -  **About Q3.** Yes, Lemma 4 provides the necessary and sufficient condition to reach an additive $\varepsilon$-approximation w.r.t. $LF'_\varepsilon$. Therefore, in the technical part of our results, we essentially try to calculate the time for the population to cover all $k$ ones for $k=0,1,\dots,n/2$ (that is, the time to witness a population $P$ with $P\cap${$x\mid |x'|_1=k$}$\neq \emptyset$ for all $k\in[0..n/2]$). Although the number of such required elements grows only linearly with the problem size $n$, the algorithm must select and keep solutions within an exponentially large objective space, making the task significantly harder than either covering the entire Pareto front in polynomial space or achieving a good approximation over a simple Pareto front in polynomial space.

---

> ### Author Response · Authors · 2025-08-04
>
> [This is meant as answer to "I think it is important to maintain consistent settings when analyzing different algorithms." Unfortunately, the system does not display it as such.] OK, we agree. We are consistent in the sense that we analyze the algorithms in the forms most commonly used in previous works. We agree that it would be nice of have analyses of SMS-EMOA and SPEA2 with crossover. Since this seems to be a shortcoming of all previous works, we are not sure if we are to blame for this. That said, we shall see what can be done here.

---

### Official Review · Reviewer_Pp8Y · 2025-06-29

**Clarity:** 3
**Significance:** 4
**Originality:** 4
**Rating:** 6
**Confidence:** 3

**Summary:**

This paper establishes, for the first time, formal runtime guarantees that explain the superior performance of widely-used Multi-Objective Evolutionary Algorithms (MOEAs)—namely NSGA-II, NSGA-III, SMS-EMOA, and SPEA2—over simpler algorithms such as GSEMO. Specifically, the authors prove that these algorithms can compute an additive ε-approximation to the Pareto front of the LF′ₑ benchmark in expected O(n² log n) fitness evaluations, while GSEMO requires exponential time.

The analysis is grounded in a general theorem (Theorem 7) applicable to any MOEA satisfying a structural diversity-preserving condition called Property A. The authors then rigorously verify that all four popular MOEAs satisfy this property under standard settings, leading to provably efficient approximation performance.

**Questions:**

1. Is the O(n² log n) bound tight? Could it be improved for specific algorithms or tie-breaking strategies, e.g., for NSGA-II with stochastic tournament or steady-state variants?

2. Are there meaningful examples of MOEAs that fail to satisfy Property A? Could the definition be relaxed, or is it necessary for achieving polynomial-time approximation?

3. Could the techniques be extended to Pareto fronts that are non-convex, disconnected, or have degeneracies? For instance, what about problems with dominance plateaus?

4. Are there insights here that suggest how to design MOEAs with provably fast approximation capabilities beyond the four studied?

5. Can your framework handle m > 2 objectives? Or does the structure of LF′ₑ and Property A critically depend on bi-objective formulation?

**Ethical Concerns:**

["NO or VERY MINOR ethics concerns only"]

**Final Justification:**

My concerns have all been resolved, so I’ll keep my score as it is.

**Limitations:**

yes

**Quality:**

4

**Strengths And Weaknesses:**

Strengths

* This is the first theoretical work to rigorously justify why popular MOEAs outperform simpler baselines like GSEMO on a challenging, established benchmark for multi-objective optimization.

* The results move MOEA theory closer to practical concerns, such as scalable approximation of large or continuous Pareto fronts, which is more relevant than full front enumeration.

* All results are stated with clear assumptions, and complete proofs are provided in the supplementary material. The use of LF′ₑ—an established and difficult benchmark—adds credibility to the claims.

Weaknesses and Limitations: While the theoretical contributions are significant, a few limitations deserve more direct attention:

* All results are derived on the LF′ₑ benchmark. While this is a well-motivated and rigorous test case, it remains a single instance of a broader problem class. The paper does not explore how the results generalize to other test suites, especially noisy, constrained, or real-valued domains.

* Although this is a theoretical paper, providing empirical illustrations (e.g., convergence rates or ε-approximation quality for various algorithms on LF′ₑ) would help contextualize the significance of the runtime bounds and provide guidance for practitioners.

* Real-world optimization often involves noise, discontinuities, or uncertainty. While the authors cite previous work analyzing NSGA-II under noise, this paper does not extend its framework or results to such settings, nor does it discuss potential obstacles.

These limitations do not undermine the core contribution, but a more explicit and detailed discussion—perhaps as a dedicated “Limitations” section—would enhance the paper’s clarity.

---

> ### Author Rebuttal · Authors · 2025-07-31
>
> Thanks a lot for the positive evaluation and nice words. Indeed, many weaknesses (which you regard as limitations) and questions are quite nice topics for our future work. Thanks again for these suggestions. Here we will give our responses one by one.
> - **(W1) Only LargeFront Benchmark.** Although LargeFront is a benchmark class, we agree that noisy or constrained cases are quite interesting. We would say the real-valued domains (or say on the continuous space) could not be an easy one, as you might know that the current evolutionary theory community mainly works on the discrete search space. We and the theory community will try our best to work more on the continuous space.
> - **(W2) Lack of empirical illustrations.** We agree that empirical validation could help, but to keep the focus and space on the theoretical contributions we chose no experiments in the main paper.
> - **(W3) Lack of discussion on noise, discontinuities, or uncertainty.** We agree that real-world problems often involve noise, uncertainty, and discontinuities. While there has been some theory work on these topic, the theoretical understanding remains limited. Thank you for the suggestion, and we will consider analyzing problems with such properties in future work.
> - **(Q1) Is $O(n\log n)$ tight? Could it be improved?** Yes, it is asymptotically tight for the algorithms analyzed in the paper. (i) For algorithms with additional mechanisms, an external-archive-based strategy that maintains a small population might achieve better runtime guarantees. (ii) Regarding the specific examples, for the steady-state variant of NSGA-II, we mentioned in the Section 4.1 (Page 5) that "the proofs are quite similar for these two variants" and "we conjecture the similar result of the steady-state variant". (iii) We also note that lower-bound results in this area are relatively difficult and often require a more delicate analysis of population dynamics. But they are important and we consider this an interesting direction for future work.
> - **(Q2) Are there meaningful examples of MOEAs that fail to satisfy Property $\mathcal{A}$? Could it be relaxed, or is it necessary?** (i) Yes, even among the four analyzed MOEAs, inappropriate parameter choices, such as an insufficient population size, can cause the algorithm fail to satisfy Property $\mathcal{A}$. (ii) As our proofs show, Property $\mathcal{A}$ is crucial for achieving polynomial-time approximation in our framework. While relaxing it may be possible, this would likely require additional mechanisms, e.g. external archives, to preserve lost solutions and maintain sufficient coverage. Exploring such extensions could be an interesting direction for future work.
> - **(Q3) Could the techniques be extended?**  Broadly speaking, our analysis relies on two main components: (i) estimating the probability that the algorithm generates necessary solutions, and (ii) understanding how the survival selection chooses among mutually non-dominated solutions. In problems with dominance plateaus or disconnected fronts, the second aspect becomes especially important. If the survival mechanism retains solutions from multiple regions despite the presence of plateaus or structural gaps, this could provide a basis for extending the analysis to more general Pareto fronts. However, formalizing this would likely require new tools or problem-specific adaptations.
> - **(Q4) Are there insights on designing MEOAs?** Thanks for this nice question, and a potential insight appears to be how the survival selection mechanism balances convergence toward the Pareto front with sufficient diversity across the objective space. Algorithms that maintain well distributed solutions while keeping the population size under control may be more likely to approximate the Pareto front efficiently. We will clarify this more clearly in our future version.
> - **(Q5) Can the framework handle more objectives?** Good question. The current LargeFront problem is a bi-objective problem. A approach similar to how prior works (e.g. [ZD24c]) generalized bi-objective benchmarks to the many-objective cases could be taken to construct an m-objective version of LargeFront. However, at this stage, we have not yet extended our analysis to $m>2$, and both the current formulation of LargeFront and Property $\mathcal{A}$ rely on the bi-objective setting. Thank the reviewer for pointing this out and we will consider analyzing the approximation for more objectives as an interesting direction for our future work.

---

> > ### Comment · Reviewer_Pp8Y · 2025-08-05
> >
> > All my concerns have been resolved.

---

> > > ### Author Response · Authors · 2025-08-05
> > > **Thank you!**
> > >
> > > Thanks a lot for the competent and encouraging review, in particular, the questions. We will use the quite summer time after the discussion phase to think more about them. Thank you for your time and brain.

---

### Official Review · Reviewer_LAZd · 2025-07-03

**Clarity:** 3
**Significance:** 4
**Originality:** 3
**Rating:** 5
**Confidence:** 3

**Summary:**

This paper attempts to provide a theoretical explanation for the widespread use of popular multi-objective evolutionary algorithms (MOEAs) by analyzing their expected runtime performance. Specifically, the authors analyze four popular MOEAs—NSGA-II, NSGA-III, SMS-EMOA, and SPEA2—on the LargeFront benchmark, and prove that each of them has an expected runtime of $O(n^2 \log n)$. The main contribution of this paper lies in establishing a theoretical runtime bound for several popular MOEAs on a benchmark specifically designed to model problems with large Pareto-optimal sets.

**Questions:**

- Could the authors clarify why decomposition-based MOEAs, such as MOEA/D, are not included in this theoretical study? A discussion on the exclusion of this important class of algorithms would help clarify the scope and limitations of the work.
- The choice of algorithmic parameters, such as the crossover rate used in the four MOEAs, requires further clarification. It would be helpful to explain whether these parameters have any impact on the theoretical analysis. For instance, it is unclear why the crossover rate is set to 0 in SMS-EMOA and SPEA2.

**Ethical Concerns:**

["NO or VERY MINOR ethics concerns only"]

**Final Justification:**

The authors have clarified why MOEA/D was not included in their analysis and have explained the relevance of the benchmark problems to real-world applications. Overall, I believe this work is significant and provides valuable theoretical insights into four popular MOEAs.

**Limitations:**

Yes

**Quality:**

3

**Strengths And Weaknesses:**

Strength
- The paper provides theoretical justification for the use of MOEAs in solving multi-objective optimization problems, contributing to a better understanding of their practical effectiveness.
- Overall, the paper is well written.

Weakness:
- The paper does not include any theoretical analysis for widely used decomposition-based MOEAs, such as MOEA/D, which limits the generality of the conclusions.
- The theoretical analysis is limited to a relatively simple benchmark (LargeFront), which may not fully reflect the complexity of real-world multi-objective problems.

---

> ### Author Rebuttal · Authors · 2025-07-31
>
> Thanks a lot for your positive comments. We now respond to the weaknesses and questions one by one.
> - **(W1 \& Q1) The analysis of the MOEA/D not included.** (1) As stated in the second paragraph in Section 1, the research line, aiming to understand why popular MOEAs are widely used, discussed NSGA-II, NSGA-III, and SMS-EMOA. Therefore, this work follows the same line and only discusses the MOEAs that do not decompose the original multi-objective problems, and this is the primary reason why the MOEA/D is not included. (2) Besides, also stated in the second paragraph in the introduction, "the simpler algorithm" (like GSEMO), compared to which we expect to see the advantages of popular MOEAs, only uses the dominance criterion for the survival selection. The popular MOEAs analyzed in this work share the same mechanism where an additional criterion is used. The comparison between these MOEAs with the simpler algorithm will lead to a clean and clear result on the effect of the additional criterion. However, the decomposition-based MOEAs like MOEA/D have a quite different framework, not good for cleanly locating the effect of a sole component. But we thank the reviewer for pointing out the important MOEA/D, and will set the approximation analysis of the MOEA/D as our interesting future work.
> - **(W2) Only limited to a simple benchmark (LargeFront).** (1) We first note that LargeFront is not one benchmark but a class of benchmarks w.r.t. $\varepsilon$ (See Definition 1, Page 2). (2) Its relation to the real-world problems. As stated in the second paragraph in Section 2.1, "the theory of evolutionary algorithms is majorly built on the discrete space" and other widely analyzed benchmarks "contain the polynomial number of Pareto (front) points". LargeFront has "exponential Pareto front points ", and is much closer to "the continuous optimization as a segment of a continuous curve in the Pareto front contains infinite points". Our results on LargeFront might shed some light on the understanding of the popular MOEAs for real-world continuous optimization.
> - **(Q2) The choice of algorithmic parameters.** (1) In general, the settings of related parameters are listed in the statement of our theorems. (2) Crossover rate. We mentioned that the crossover rate is set to 0 for the SMS-EMOA and the SPEA2, when we discussed how these two algorithms fit into our framework in Algorithm 1. This setting follows the previous theoretical studies on these algorithms, see [BZLQ23,ZD24c,RBLQ24] in the Reference section. Indeed, our general theorem (Theorem 7) holds for any fixed crossover rate $p_c\in[0,1)$ (see the statement of Theorem 7). As demonstrated in our proofs, the crossover rate does not influence the asymptotic runtime bounds we establish. Thanks for raising this question, and we will make it clearer in our future version.

---

> > ### Comment · Reviewer_LAZd · 2025-08-05
> >
> > I have read the authors' responses, as well as their discussion with the other reviewers. Based on these responses, I think the authors have addressed my questions. Therefore, I will raise my score to support the acceptance of this work.

---

> > > ### Author Response · Authors · 2025-08-05
> > > **Thank you!**
> > >
> > > Thanks a lot for reading our rebuttal and for the fruitful discussion. We really appreciate the time and energy you spent on our paper. Thank you!

---

### Decision · Program_Chairs · 2025-09-17

**Decision:**

Accept (poster)

**Comment:**

This paper provides formal complexity guarantees for a range of well-known Multi-Objective Evolutionary Algorithms.
It is shown that these algorithms compute an additive approximations to the Pareto set of benchmark datasets
in expected O(n^2 log n) time. On the other hand, other simpler methods requires exponential time for the same task. After a robust discussion, the consensus of the reviewers is that this is an interesting contribution that merits acceptance.